# Oxidative Depolymerization of Cellulolytic Enzyme Lignin over Silicotungvanadium Polyoxometalates

**DOI:** 10.3390/polym11030564

**Published:** 2019-03-26

**Authors:** Wenbiao Xu, Xiangyu Li, Junyou Shi

**Affiliations:** 1Key Laboratory of Bio-based Material Science and Technology of Ministry of Education, Northeast Forestry University, Harbin 150040, China; wenbiao.xu@mail.utoronto.ca; 2Jilin Provincial Key Laboratory of Wooden Materials Science and Engineering, Beihua University, Binjiang East Road, Jilin 132000, China; lxy@beihua.edu.cn

**Keywords:** cellulolytic enzyme lignin, polyoxometalates, oxidation, aromatic compounds

## Abstract

The aim of this study was to explore the catalytic performance of the oxidative depolymerization of enzymatic hydrolysis lignin from cellulosic ethanol fermentation residue by different vanadium substituted Keggin-type polyoxometalates (K_5_[SiVW_11_O_40_], K_6_[SiV_2_W_10_O_40_], and K_6_H[SiV_3_W_9_O_40_]). Depolymerized products were analyzed by gel permeation chromatography (GPC), gas chromatography–mass spectrometer (GC/MS), and two-dimensional heteronuclear single quantum coherence nuclear magnetic resonance (2D HSQC NMR) analysis. All catalysts showed an effective catalytic activity. The best result, concerning the lignin conversion and lignin oil production, was obtained by K_6_[SiV_2_W_10_O_40_], and the highest yield of oxidative depolymerization products of 53 wt % was achieved and the main products were monomer aromatic compounds. The HSQC demonstrated that the catalysts were very effective in breaking the β-O-4 structure, the dominant linkage in lignin, and the GPC analysis demonstrated that the molecular of lignin was declined significantly. These results demonstrate the vanadium substituted silicotungstic polyoxometalates were of highly active and stable catalysts for lignin conversion, and this strategy has the potential to be applicable for production of value-added chemicals from biorefinery lignin.

## 1. Introduction

Lignin is a heterogeneous renewable biopolymer and the only feedstock in nature with high carbon content and high aromaticity [1]. It is chemically and physically interlaced with cellulose and hemicelluloses in the plant cell walls. Its complex chemical structure and stable chemical properties make most lignin degradation a highly challenging work [2,3]. In a typical biorefinery, the natural lignin is structurally modified under the traditional acid or high temperature fraction or pre-treatment, leading to the irreversible condensation that dramatically affects its further catalytic valorization [2,4,5]. At present, lignin as a raw material for the production of value-added products as an alternative to fossil derived chemicals has attracted ever increasing attention during the past decade, and it is also apparently the most encouraging transformation to high-value utilization of lignin [6]. The processes for conversion lignin can be broadly classified into base or acid depolymerization [7], pyrolysis [8,9,10], hydroprocessing [11,12], oxidation [13,14,15], and other depolymerization processes. Among the above catalytic transformation strategies, oxidative depolymerization is a very promising valorization method and represents various advantages, since it focuses on production of highly functionalized platform aromatic compounds which can be used as starting materials in chemical and pharmaceutical industries [16,17,18].

Polyoxometalates (POMs) are composed of metals (usually transition metals) and oxygen, which can be divided into homonuclear and heteronuclear compounds. Varying the structure and composition of the POM affects the physical and chemical (especially acidity and redox potential) properties [19]. Polyoxometalates have previously been used instead of chlorine-based bleaching chemicals in the delignification of pulp, where they were employed as delignification agents to remove the residual lignin with environmentally benign oxidation processes [14]. Von Rohr and co-workers extensively studied the catalytic transformation of lignin over commercially available H_3_PMo_12_O_40_, as its redox potential was high enough to oxidize lignin and low enough to be reoxidized by oxygen [20,21]. Moreover, they observed that low molecular alcohols such as methanol could effectively prevent the repolymerization of lignin fragments [22]. Although some achievements have been made in the catalytic depolymerization of industrial lignin or lignin model compounds catalyzed by POMs [14,23,24,25,26], these researches are restricted to the polyoxometalates delignification or simple model compounds. Very little information is available concerning chemical production and the chemical structure changes of the lignin macromolecule at the molecular level [14,20].

The cellulolytic enzyme lignin (CEL) is the by-product of the bioethanol residual using corn stalk as feedstock. Unlike the common bioconversion routes for cellulose and hemicellulose utilization, lignin is non-fermentable to microorganisms during the bioethanol production and currently utilized for low-quality materials or burned as fuel [27]. The valorization of the large amount of generated CEL is one of the crucial considerations for the sustainable and economical for bioethanol production [28]. Our group previously activated the CEL by laccase and applied it as formaldehyde-free plywood adhesive, a very well-known lignin application [29]. In fact, the enzyme process occurs at mild conditions, and the original lignin structure is basically kept that can be further utilized to produce valuable chemicals. Liu’s group described that the catalytic transformation of cornstalk lignin to aromatic aldehydes, using perovskite-type oxidative catalysts of La and Mn [30] or Co [31]. This strategy has been proved to be somewhat successful, but the isolated yield of products is significantly less than other depolymerization methods.

Herein, several vanadium substituted Keggin-type silicotungstic polyoxometalates were prepared and applied to the catalytic depolymerization of CEL to low molecular aromatics. A comparison between vanadium content and catalytic performance of different catalysts was presented and the possible depolymerization mechanism was discussed. To the best of our knowledge, there is no study on the catalytic performance of silicotungvanadium polyoxometalates on the CEL, especially for aromatic compound production.

## 2. Materials and Methods

### 2.1. Materials

All commercially available chemicals were used as received unless otherwise noted. The bio-refinery residue was supplied by Henan Tianguan Group Co., Ltd., Nanyang, China. The residue was air-dried at room temperature to equilibrium moisture content and the purification of CEL were prepared as the following method: 5.0 g cellulosic ethanol fermentation residue was added to 40 mL 3% NaOH aqueous solution. This mixture was stirred at 65 °C for 90 min. After filtration, the lignin solution was collected, and 2% H_2_SO_4_ was added to adjust pH to about 3. Then, the mixture was heated at 70 °C for 30 min. After standing for 24 h, the mixture was separated by centrifuge. The precipitate was washed three times with water and then evaporated to dry at room temperature. The dried residue was dissolved in 10 mL dioxane and precipitated by the ether (20 mL). After filtration and desiccation, the purified CEL was obtained and used in subsequent experiments without further processing. The compositional analysis was performed by the standard NREL laboratory analytical procedure (NREL/TP-510-42618). The ash content of the CEL was obtained gravimetrically after calcined in a muffle furnace at 575 °C. This purified CEL contained 90.6% acid-insoluble lignin, 3.8% acid-soluble lignin, 0.08% arabinose, 0.11% glucose, and 0.09% xylose. The ash and moisture contents of the CEL were 1.2% and 4.12%, respectively.

### 2.2. Preparation and Characterization of the Catalysts

The vanadium substituted silicotungstic polyoxometalates K_5_[SiVW_11_O_40_], K_6_[SiV_2_W_10_O_40_], and K_6_H[SiV_3_W_9_O_40_] were prepared according to the literature method [32,33], with some modifications. FTIR spectroscopy was recorded in KBr discs on a Nicolet Magna 560 IR spectrometer. Scanning electron microscope (SEM) was carried out by a FEI-Quanta 200 (Madison, America) equipped with energy dispersive X-ray spectrometer (EDX) analyses at 100 kV, EDX elemental mapping were operated at 200 kV. The redox potential was measured by cyclic voltammetry (CV) on a CS Corrtest CHI760E electrochemical workstation (Shanghai, China) equipped with a carbon paste electrode and saturated calomel as the reference electrode in an electrolyte of 0.1 M sulfuric acid solution.

The FTIR spectra of different vanadium substituted silicotungstic polyoxometalates were investigated. As shown in Figure 1, there were four characteristic peaks at 1008 cm^−1^ (*vas* Si-O_a_, internal oxygen connecting Si and W), 962 cm^−1^ (*vas* W-O_d_, terminal oxygen bonding to W atom), 896 cm^−1^ (*vas* W-O_b_, edge-sharing oxygen connecting W), and 755cm^−1^ (*vas* W-O_c_, corner-sharing oxygen connecting WO_6_ units) ranged from 600 to 1100 cm^−1^, which were attributed to Keggin structural vibrations. Figure 1 also shows the XRD patterns of these three silicotungvanadium polyoxometalates with characteristic diffraction peaks at 8.5°, 10.3°, 25.8° and 34.6°, which were attributed to body-centered cubic secondary structure of Keggin anion [34].

Figure 2 shows the SEM images of K_5_[SiVW_11_O_40_], K_6_[SiV_2_W_10_O_40_], and K_6_H[SiV_3_W_9_O_40_]. These catalysts displayed well-shaped crystalline particles, especially for the one and two vanadium substituted POMs. With the increase of vanadium, the size of these particles gradually decreased, which was due to the ionic radius of tungsten being larger than vanadium. Moreover, EDX was employed to determine the elemental composition of these catalysts. The predominant components were all found to be Si, W, V, and O, in accordance with the proportion of theoretical stoichiometry. CV-potential curves of vanadium-substituted polyoxometalates was shown in Figure 3. These results showed that the introduction of vanadium can alter the redox properties of POMs, and the structural and property performance differences caused by different vanadium substitutions will play an important role in lignin transformation.

### 2.3. Catalytic Decomposition of the CEL

In a typical reaction protocol, 0.25 g of purified CEL, 0.1 mmol POMs, 8 mL CH_3_OH, and 2 mL H_2_O were charged into a 10 mL stainless autoclave. The reactor was sealed and purged three times with oxygen, pressurized to 2.0 MPa O_2_ at room temperature and then heated to the desired temperature with stirring at 500 rpm. After the reaction, the autoclave was cooled to room temperature and depressurized. The reaction mixture was filtered, and the insoluble fraction was washed with methanol, the solution was totally transferred to a round-bottom flask and the methanol was removed under vacuum condition. The obtained mixture was extracted with dichloromethane (DCM) three times (10 mL × 3) and the organic layer was dried with anhydrous MgSO_4_ then evaporated under reduced pressure. The lignin conversion and lignin oil yields were calculated using the following equations:$$\mathrm{Lignin}\;\mathrm{conversion}\;\left(\%\right)=\frac{w\;\left(initial\;lignin\right)-w\;\left(lignin\;residue\right)}{w\;\left(initial\;lignin\right)}\;\times 100\%.\;$$
$$\mathrm{Lignin}\;\mathrm{oil}\;\mathrm{yield}\;\left(\%\right)=\frac{w\;\left(DCM\;soluble\right)}{w\;\left(initial\;lignin\right)}\;\times 100\%$$

### 2.4. Product Analysis

The HSQC NMR spectra was carried out at 25 °C on a Bruker-Advance III 500 MHz spectrometer (Karlsruhe, Germany) equipped with a z-gradient double resonance probe. A 100 mg sample was dissolved in 0.4 mL of DMSO-*d*_6_ and 0.1 mL pyridine-*d*_5_. The experiment parameters were described previously [35]. The depolymerized aromatic compounds were also analyzed by GC/MS using a Shimadzu GC/MS-QP2010SE (Kyoto, Japan) equipped with an SH-Rxi-5Sil capillary column, and the parameters were described previously [36]. Gel permeation chromatography (GPC) analysis was conducted using an Agilent Infinity 1260 HPLC (Palo Alto, CA, USA) equipped with a Refractive Index Detector and an Agilent PLgel MIXED C column at 40 °C using tetrahydrofuran (THF) as the mobile phase and a flow rate of 1 mL/min. Before the GPC experiment, the CEL and lignin oil products were acetylated as shown briefly below: CEL (10 mg) or lignin oil products (10 mg) were treated with a 1:1 mixture of acetic anhydride and pyridine (1.0 mL) at room temperature for 48 h. The acetylated products were extracted by ethyl acetate and the organic layer was washed with brine and dried over anhydrous MgSO_4_. After evaporation under vacuum, the obtained acetylation products were dissolved in THF (2 mg mL^−1^) and filtered over a 0.45 μm syringe filter prior to injection.

## 3. Results and Discussion

### 3.1. Catalytic Oxidation Depolymerization of CEL

The catalytic efficiency of different vanadium substituted silicotungstic polyoxometalates were evaluated by the measurement of lignin conversion and lignin oil yield. As shown in Figure 4, the K_6_[SiV_2_W_10_O_40_] with two vanadium atoms exhibited both highest lignin conversion at 87% and lignin oil yield at 53%, respectively. As compared to the silicotungvanadium POMs, the control reaction without catalysts and with a vanadium-free POM K_8_[SiW_11_O_39_] showed both the low lignin conversion and lignin oil yield. This result reflected that the number of vanadium had significant effect on the catalytic transformation of lignin. The use of vanadium-substituted polyoxometalates with strong redox ability enhances the hydrogen liberation and the liberated hydrogen transfer to the reaction intermediates. As a consequence, the lignin conversion and products yields were enhanced. In general, the redox ability of POMs increases with the number of substitutions of vanadium, contributing to the increase of lignin conversion. During these reactions, K_6_[SiV_2_W_10_O_40_] performed higher efficiency than K_5_[SiVW_11_O_40_]. This may be attributed to the higher redox potential introduced by vanadium. However, the K_6_H[SiV_3_W_9_O_40_] with more vanadium atoms did not follow this rule. The possible explanation was that the lignin was over oxidized to carbon dioxide and water, and the lignin fragments were recombined during the reaction. The detail information about the oxidative reaction was characterized by GPC, HSQC and GC/MS as below.

To confirm the depolymerized products, the molecular weight distribution was investigated by GPC (Figure 5). As shown in Figure 5, the parent lignin was essentially insoluble in THF (presumably due to high molecular weight), whereas the post-reaction oils were all significantly soluble. GPC profiles of the CEL and depolymerized products catalyzed by the different number of vanadium substituted POMs suggested the molecular weight was significantly decreased in average molecular weight (*M_W_* 1300 g mol^−1^) relative to the raw CEL (*M_W_* 4200 g mol^−1^). Further insight into the structural changes were analyzed using two-dimensional heteronuclear single quantum coherence nuclear magnetic resonance (2D HSQC NMR) which was particularly found to be helpful to monitor the changes of lignin structure after oxidative decomposition reaction. The 2D HSQC NMR spectra of CEL mainly separated for three regions, namely the aliphatic region (generally δ_C_/δ_H_ 0–50/0–2.5 ppm), side chain region (generally δ_C_/δ_H_ 50–100/2.5–6.5 ppm), and aromatic region (generally δ_C_/δ_H_ 100–140/6.5–8 ppm) [37,38]. The inter-unit linkages and structural units could be characterized by the side chain region and aromatic region respectively. The aliphatic region was not related with the lignin structure, therefore, it was not discussed here. The detailed information is presented in Figure 6, the assignments of lignin samples were based on recent studies [1,39,40]. By calculating the Cα integral values of the HSQC signals, the relative percentages of the main three inter-unit linkages, namely, β-O-4 (A), β-5(B), and β-β(C) were 80%, 14%, and 6% respectively (Figure 6). After the reaction, as we can see from Figure 6b,c,d, the signals of A_α_ (δ_C_/δ_H_ 72.3/5.1 ppm) and A_β_ (δ_C_/δ_H_ 86.7/4.3 ppm) attributed to the benzylic hydroxyl and secondary alkyl protons, respectively, were vanished completely. This result indicated the silicotungvanadium POMs were effective in the cleavage of C-O-C linkages in CEL under this reaction condition. Except the β-O-4 linkage, the signals for the β-5 and β-β inter-unit linkages disappeared as well. For the aromatic area, we can also observe that the Cα hydroxyl group oxidized syringyl type units generated obviously after the reaction, and the syringyl to guaiacyl ratio after hydrolysis was increased slightly but remained basically the same as the unreacted CEL. These results were in accordance with the analysis detected by GC/MS.

### 3.2. Effect of the Reaction Time and Temperature

The effects of reaction conditions including reaction time and reaction temperature on lignin decomposition are shown in Figure 7. When the reaction time was prolonged from 2 h to 3 h, the lignin conversion yield and oily products yield changed from 74% and 48% to 87% and 53%, respectively. Prolong the reaction time to 6 h, the lignin conversion yield and oily products yield decreased to 83% and 44%, which was probably due to the slight repolymerization reactions of intermediate lignin fragments. These condensation reactions of lignin using POMs as catalysts have been well studied in previous reports [14]. A similar volcano-shaped trend was also observed at different reaction temperatures. Increase in the reaction temperature led to an increase of lignin conversion yield and oily products yield to 150 °C. When temperature was further increased to 190 °C, the lignin oily products yield was decreased from 52.7 wt % at 150 °C to 43.6 wt % at 190 °C. In brief, with the progress of the reaction, higher reaction temperature and longer reaction time would facilitate the depolymerization reaction, but the undesirable condensation reactions occurred in the final stage of reaction, thus inhibiting the efficient degradation of lignin.

### 3.3. Depolymerization Products Analysis

Although GPC and HSQC are useful in characterization of molecular weight distribution and structure changes, respectively, they are not powerful at identifying the individual depolymerized products. The purpose of this study is to produce monomer aromatic compounds, so the GC/MS analysis was also employed on the reaction mixture catalysed by K_6_[SiV_2_W_10_O_40_]. As shown in Table 1, the 2-methoxy-4-vinylphenol and vanillin, acetosyringone, and 4-hydroxybenzaldehyde and 4-(methoxycarbonyl)-phenol are the most abundant aromatic compounds in the reaction mixture, derived from the G, S and H units of CEL, respectively. Among these aromatics, the vanillin is a very important commercial compound used in food, cosmetic and pharmaceutical industries [41], and the functionality around the aromatic ring allows it an adaptable platform compounds to synthesize numerous polymers [42]. The other compounds like acetosyringone can also be used in medicine, perfume, pesticide chemistry and organic synthesis industry.

### 3.4. Recycling Experiments

Reusability of the catalyst is one of the most important performance requirements in practical applications. During the lignin depolymerization process, the most important properties of POMs are their re-oxidation with oxygen, and the deoxidized POMs can be used for the next catalytic circulation. Considering the silicotungvanadium polyoxometalates are homogeneous catalysts, the aqueous phase of the product extraction step containing all the reacted catalysts was evaporated to dryness by water bath and was then subjected to next reaction. As shown in Figure 8, after five cycles under the optimizing conditions, the lignin oil yield had almost no significant changes with only 6% decrease. The results suggested the vanadium substituted silicotungstic polyoxometalates was stable under such conditions and could be reused without significant loss of its catalytic activity.

### 3.5. Possible Mechanisms of Lignin Depolymerization

According to the chemical properties of the substrate, there are main three pathways during the oxidative reaction when using POMs as catalysts. Namely, electron-transfer, O-transfer, and radical-transfer [43]. Based on the above discussion on the products and structure changes before and after reaction, a mechanism is proposed for describing the lignin depolymerization over silicotungvanadium polyoxometalates. As shown in Figure 9, the POMs ions undergo repeated cycles of reduction and re-oxidation, which depends on whether the redox potential of POMs is high enough to oxidize the lignin substrate and low enough to be oxidized by oxygen [20,26]. At step 1, the POMs with a redox electrochemical potential oxidized the lignin substrate and the POMs was reduced. During the oxidation, the benzyl hydroxyl was selectively oxidized into the corresponding carbonyl, which could lower the bond energy of the C-O bond and make the β-O-4 linkages easier to fracture [2]. Chemoselective oxidation of the benzylic hydroxyl groups in lignin polymer is of great significance in the lignin oxidative depolymerization, and it represents an applicable way for facilitating the lignin depolymerization [44,45]. After the step 1, the insoluble macromolecular lignin turn into the soluble lignin_ox_, then the lignin^ox^ undergo C–C and/or C–O cleavage, resulting in the forming of low molecular of aromatic compounds and partial oxidation of the lignin fragments to CO_2_ and H_2_O. Finally, the catalytic cycle is completed by the re-oxidation of POM_red_ to POM_ox_ form using O_2_ (Step 2). The regenerated POMs could be reused in the next catalytic circulation.

## 4. Conclusions

In summary, we developed an effective degradation strategy for biorefinery CEL in the production of aromatic compounds. Our chemical conversion uses reuseable polyoxometalates to transform CEL into valuable aromatic compounds with an acceptable conversion and yield. The appropriate substituted vanadium number with high catalytic activity could easily oxidise lignin substrate followed by homolytic cleavage of C—C and C—O bonds. The catalyst could be reused five times without obvious loss of catalytic activity. Thus, the strategy of vanadium substituted Keggin-type silicotungstic polyoxometalates was evaluated as a potential way for the valorization of biorefinery lignin.

## Figures and Tables

**Figure 1 polymers-11-00564-f001:**
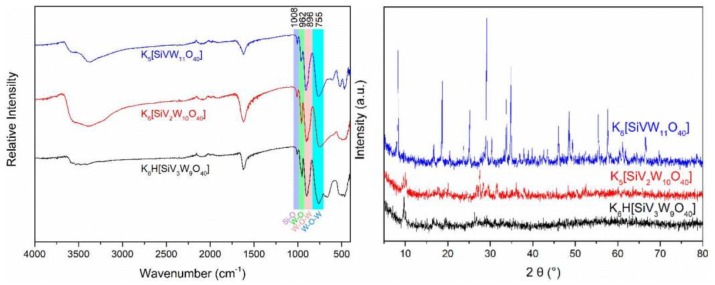
FTIR (**left**) and XRD (**right**) spectrum of the silicotungvanadium polyoxometalates.

**Figure 2 polymers-11-00564-f002:**
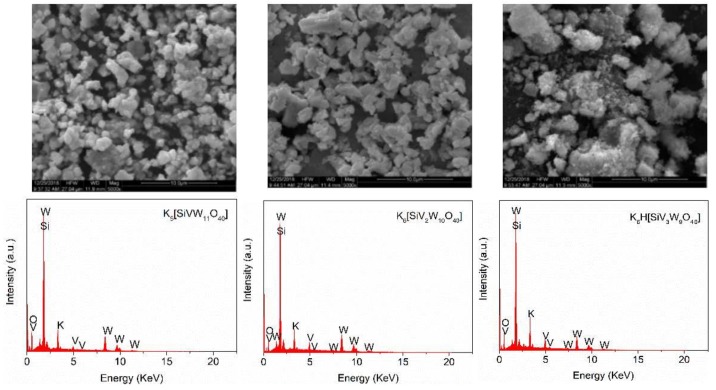
SEM micrographs (up) and EDX-spectrum (down) of the catalysts K_5_[SiVW_11_O_40_], K_6_[SiV_2_W_10_O_40_], and K_6_H[SiV_3_W_9_O_40_] from left to right respectively.

**Figure 3 polymers-11-00564-f003:**
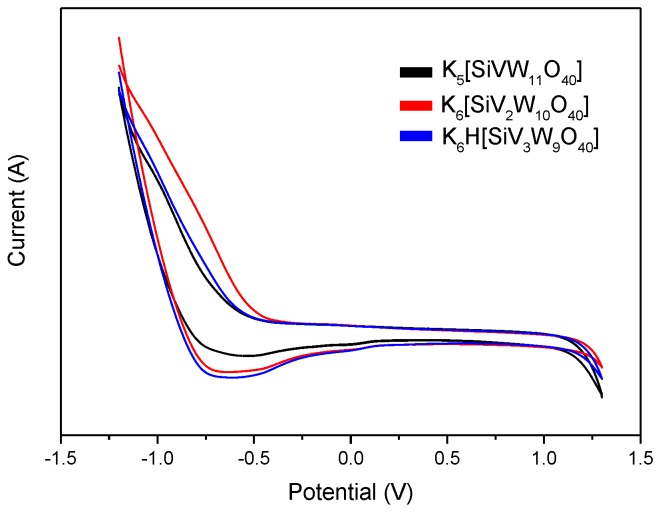
CV-potential curves of vanadium-substituted polyoxometalates.

**Figure 4 polymers-11-00564-f004:**
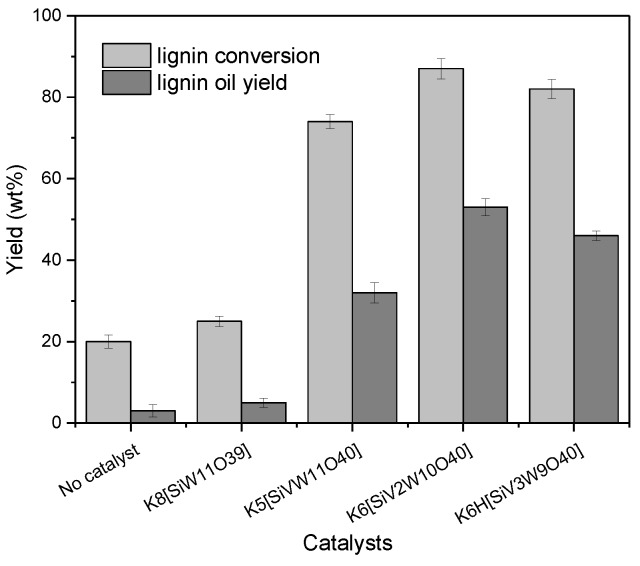
The product yield of the catalytic oxidation of cellulolytic enzyme lignin by different silicotungvanadium polyoxometalates. Reaction conditions: Lignin 0.25 g, catalysts 0.1 mmol, CH_3_OH/H_2_O = 8:2 10 mL, O_2_ 2 MPa, 150 °C, 3 h, error bars represent standard deviations from three replicates.

**Figure 5 polymers-11-00564-f005:**
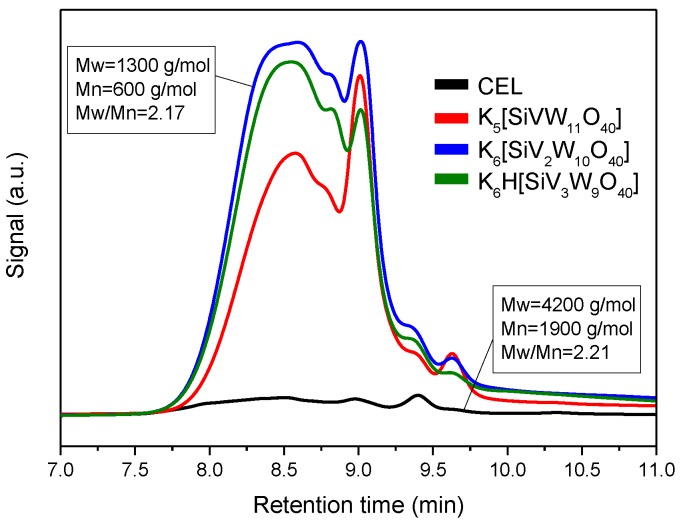
Gel permeation chromatograms (GPC) of the lignin and oily products, comparing the molecular distribution in function of catalysts. Reaction conditions: Lignin 0.25 g, catalyst 0.1 mmol, CH_3_OH/H_2_O = 8:2 10 mL, O_2_ 2 MPa, 150 °C, 3 h.

**Figure 6 polymers-11-00564-f006:**
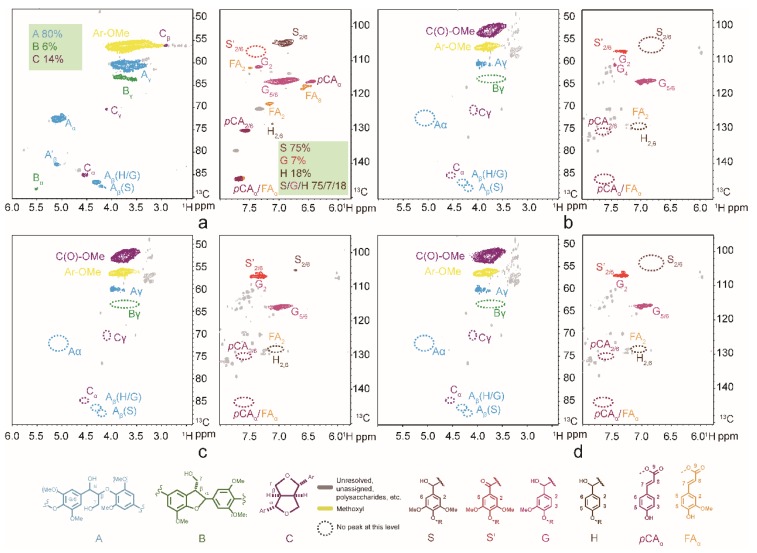
Expanded side-chain region and aromatic region of the two-dimensional heteronuclear single quantum coherence nuclear magnetic resonance spectra of cellulolytic enzyme lignin (CEL) (**a**) and lignin oily products (**b**) catalyzed by K_5_[SiVW_11_O_40_]; (**c**) catalyzed by K_6_[SiV_2_W_10_O_40_]; (**d**) catalyzed by K_6_H[SiV_3_W_9_O_40_]) after oxidative depolymerization reaction in DMSO-*d*_6_.

**Figure 7 polymers-11-00564-f007:**
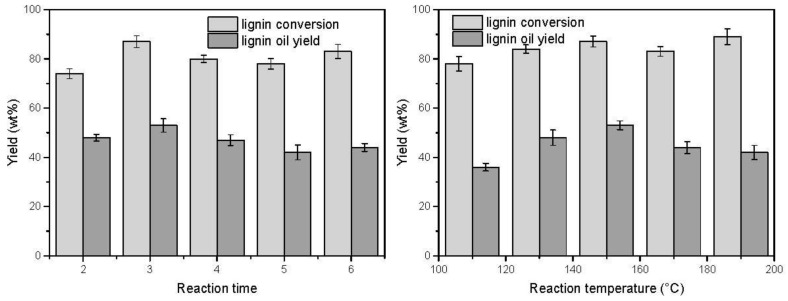
The product yield of the catalytic oxidation of CEL at different reaction times and temperatures. Reaction for left: Lignin 0.25 g, K_6_[SiV_2_W_10_O_40_] 0.1 mmol, CH_3_OH/H_2_O = 8:2 10 mL, O_2_ 2 MPa, 150 °C. Reaction for right: Lignin 0.25 g, K_6_[SiV_2_W_10_O_40_] 0.1 mmol, CH_3_OH/H_2_O = 8:2 10 mL, O_2_ 2 MPa, 3 h.

**Figure 8 polymers-11-00564-f008:**
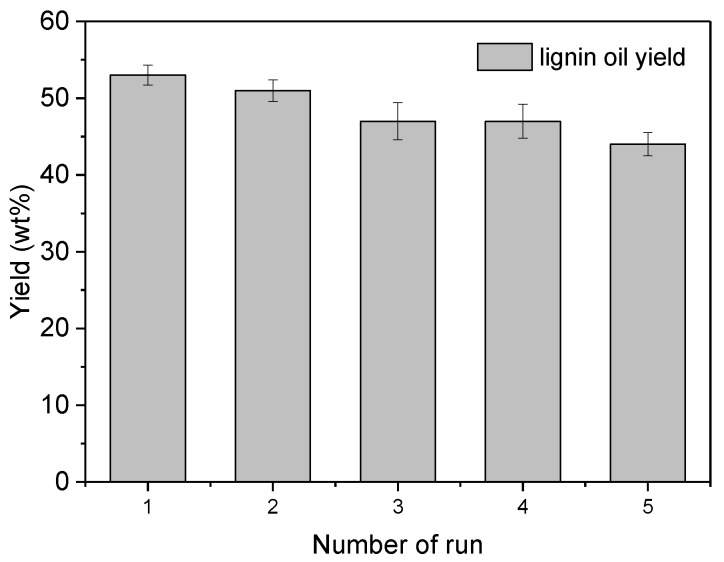
Repeated uses of K_6_[SiV_2_W_10_O_40_] for CEL depolymerization (reaction conditions: Lignin 0.25 g, K_6_[SiV_2_W_10_O_40_] 0.1 mmol, CH_3_OH/H_2_O = 8:2 10 mL, O_2_ 2 MPa, 150 °C, 3 h, error bars represent standard deviations from three replicates).

**Figure 9 polymers-11-00564-f009:**
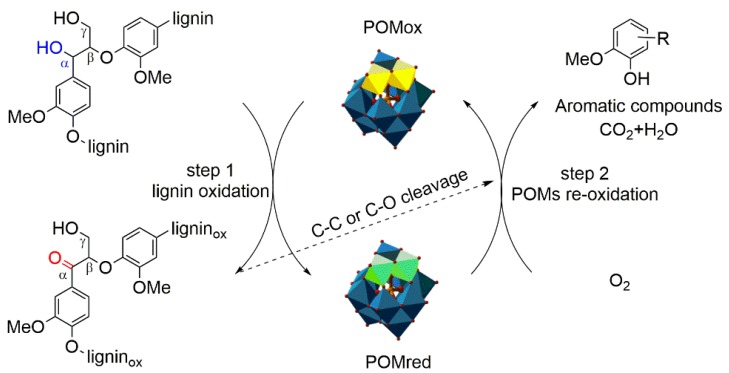
Proposed reaction mechanism for the silicotungvanadium polyoxometalates oxidation depolymerization of CEL. (The yellow and green color represent the substituted vanadium atom).

**Table 1 polymers-11-00564-t001:** The main phenolic compounds in the reaction mixture detected by GC/MS (reaction conditions: Lignin 0.25 g, K_6_[SiV_2_W_10_O_40_] 0.1 mmol, CH_3_OH/H_2_O = 8:2 10 mL, O_2_ 2 MPa, 150 °C, 3 h).

Entry	Retention Time	Phenolic Compounds	Structure	Yield (%)
1	7.569	2-Methoxy-4-vinylphenol	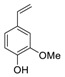	3.43
2	8.363	4-Hydroxybenzaldehyde	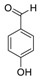	0.39
3	9.040	Vanillin	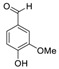	5.46
4	9.975	4-(Methoxycarbonyl)phenol	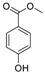	0.78
5	15.145	Acetosyringone	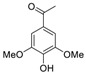	0.18

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
