# Peer review of "Oxidative Depolymerization of Cellulolytic Enzyme Lignin over Silicotungvanadium Polyoxometalates"

_polymers, 2019, doi:10.3390/polym11030564_

Round 1

Reviewer 1 Report

This paper describes the depolymerization of cellulolytic enzyme lignin from an ethanol production facility using vanadium-doped polyoxometalate catalysts. The authors show that doping an intermediate level of V into the POM structure is likely optimal, and demonstrate that the catalyst is capable of cleaving b-O-4 bonds as well as (remarkably) b-5 and b-b C-C linkages. The results are interesting and should be published in Polymers, but only after major revisions as there are numerous shortcomings that need to be remedied before publication.

The main issue is that there are two control reactions missing—the authors need to test the conversion of CEL without catalyst, and with a vanadium-free POM to really know the effect of the catalyst, and specifically, the vanadium incorporation into the catalyst. The other main issue is that the authors need to report yields of their identified monomers, rather than just identification of them.

There are also several other minor issues:

Throughout: the authors should proofread the manuscript for grammatical errors.

Pg 2, line 58: lignin is non-fermentable to micro-organisms: this statement is incomplete, as there is a large body of research on lignin-degrading micro-organisms, including white-rot fungi and Pseudomonas putida.

Pg 2, lines 65-68: Perovskites are not POMs; if these oxidation catalysts are to be discussed here, so should many other types (a discussion which would strengthen the introduction, for sure). In addition to the references already included, there are a couple more papers from Voitl and von Rohr :

T. Voitl , M. V. Nagel and P. R. von Rohr , Holzforschung, 2010, 64 , 13 -19;

H. Werhan , J. M. Mir , T. Voitl and P. R. von Rohr , Holzforschung, 2011, 65 , 703 -709

Also, the authors should motivate their selection of this particular type of POM—what hypothesis are they testing in these experiments? It seems to be that substituting V into a POM structure will increase the POM’s activity toward lignin oxidation, but they don’t explicitly say that or explain why they think so.

Pg 2, line 86: have the authors done any characterization of this lignin stream? What is the composition with respect to residual carbohydrates and ash?

Pg 4, lines 150-151: is there really a difference between the V2 and V3 results? If the results shown in Figure 3 are claimed to be different, there should be some error bars on them.  Also, what was the time and temperatures used to generate the data in Figure 3?

Figure 4: It would be helpful to explain that the figure is showing only THF-soluble products, indicating that the parent lignin is essentially insoluble in THF (presumably due to high MW), whereas the post-reaction oils are all significantly soluble. An indication of approximate MW in the figure would also be helpful.

Figure 6 (first one): The reaction conditions that were held constant in these experiments should be denoted (e.g., what temperature was the left-side panel run at, and what time was the right-side panel run at?)

Table 1: The field has really moved beyond simple identification of products by GCMS. The authors need to provide the yield of monomeric products.

Pg 8, line 225: it looks like the decrease in oil yield is about 10% absolute (more like 15-20% relative), so calling it a “stable” catalyst seems like a bit of a stretch. Some post-reaction catalyst characterization would help evaluate changes to the catalyst. Also, Figure 7 is labeled as Figure 6.

Author Response

List of Responses

Dear Editors and Reviewers:

Thank you for your letter and for the reviewers’ comments concerning our manuscript entitled “Oxidative depolymerization of cellulolytic enzyme lignin over silicotungvanadium polyoxometalates” (ID: 443754). Those comments are all valuable and very helpful for revising and improving our paper, as well as the important guiding significance to our researches. We have studied comments carefully and have made correction which we hope meet with approval. Revised portion are marked using the “Track Changes” function in Microsoft Word. The main corrections in the paper and the responds to the reviewer’s comments are as flowing:

The main issue is that there are two control reactions missing—the authors need to test the conversion of CEL without catalyst, and with a vanadium-free POM to really know the effect of the catalyst, and specifically, the vanadium incorporation into the catalyst. The other main issue is that the authors need to report yields of their identified monomers, rather than just identification of them.

Response: Considering the reviewer’s suggestion, we have supplemented the corresponding experiments, including the investigation of the conversion of CEL without catalyst, and with a vanadium-free POM K8[SiW11O39] (as shown in figure 4). The results of the product yield by different POMs were also updated in the revised manuscript (as shown in table 1).

Figure 4. The product yield of the catalytic oxidation of cellulolytic enzyme lignin by different silicotungvanadium polyoxometalates. Reaction conditions: Lignin 0.25 g, catalysts 0.1 mmol, CH3OH/H2O=8:2 10 ml, O2 2 MPa, 150 °C, 3 h, error bars represent standard deviations from three replicates.

Table 1 The main phenolic compounds in the reaction mixture detected by GC/MS (reaction conditions: Lignin 0.25 g, K6[SiV2W10O40] 0.1 mmol, CH3OH/H2O=8:2 10 ml, O2 2 MPa, 150 °C, 3 h).

Entry

Retention   time

Phenolic   compounds

structure

Yield (%)

1

7.569

2-Methoxy-4-vinylphenol

3.43

2

8.363

4-Hydroxybenzaldehyde

0.39

3

9.040

Vanillin

5.46

4

9.975

4-(Methoxycarbonyl)phenol

0.78

5

15.145

Acetosyringone

0.18

There are also several other minor issues:

Throughout: the authors should proofread the manuscript for grammatical errors.

Response: We checked the grammatical errors carefully and the main corrections as flowing:

1. Line 15, the statements of “observed” were corrected as “analyzed”.

2. Line 32, “and” was deleted.

3. Line 64, the statements of “economically” were corrected as “economical”.

5. Line 190, “during the catalytic oxidative reactions” was deleted.

6. Line 191, the statements of “obtained” were corrected as “analyzed”.

7. Line 192, the statements of “in the structure” were corrected as “of lignin structure”.

Pg 2, line 58: lignin is non-fermentable to micro-organisms: this statement is incomplete, as there is a large body of research on lignin-degrading micro-organisms, including white-rot fungi and Pseudomonas putida.

Response: We intended to express that lignin is non-fermentable to micro-organisms during the bioethanol production. Considering the possibility of misunderstanding, we have made a modification in the manuscript.

Pg 2, lines 65-68: Perovskites are not POMs; if these oxidation catalysts are to be discussed here, so should many other types (a discussion which would strengthen the introduction, for sure). In addition to the references already included, there are a couple more papers from Voitl and von Rohr :

T. Voitl , M. V. Nagel and P. R. von Rohr , Holzforschung, 2010, 64 , 13 -19;

H. Werhan , J. M. Mir , T. Voitl and P. R. von Rohr , Holzforschung, 2011, 65 , 703 -709

Response: We are very sorry for our negligence of understanding about perovskite. We corrected the error and added the references mentioned in the comments. Perovskites were described as oxidative catalysts, and the references were added as “Von Rohr and co-workers extensively studied the catalytic transformation of lignin over commercially available H3PMo12O40, as its redox potential was high enough to oxidize lignin and low enough to be reoxidized by oxygen[23, 24]. Moreover, they observed the low molecular alcohols such as methanol could effectively prevent the repolymerization of lignin fragments.”

Also, the authors should motivate their selection of this particular type of POM—what hypothesis are they testing in these experiments? It seems to be that substituting V into a POM structure will increase the POM’s activity toward lignin oxidation, but they don’t explicitly say that or explain why they think so.

Response: We believe that the redox potential of the catalysts plays an important role in lignin conversion. So, we added the CV-potential curves of vanadium-substituted polyoxometalates in the revised manuscript (as shown in figure L1). Theoretically, POMs, with a reduction potential sufficiently high to oxidize lignin but low enough for reoxidation with oxygen, are capable of transferring electrons from lignin to oxygen (E0lignin < E0POM < E0O). The use of vanadium-substituted polyoxometalates with strong redox ability, enhances the hydrogen liberation and the liberated hydrogen transfer to the reaction intermediates. As a consequence, the lignin conversion and products yields were enhanced. In general, the redox ability of POMs increases with the number of substitutions of vanadium, contributing to the increase of lignin conversion. However, the excessive redox ability will lead to the deep oxidation of lignin to carbon dioxide and water. This phenomenon can be explained by the fact that the lignin conversion catalyzed by V2 and V3 were almost constant but the product yield catalyzed by V3 was reduced (as shown in Figure 4). Moreover, the redox properties of the POMs will be further investigated by H2-TRP and XPS experiments in our future research.

Figure L1. CV-potential curves of vanadium-substituted polyoxometalates.

Pg 2, line 86: have the authors done any characterization of this lignin stream? What is the composition with respect to residual carbohydrates and ash?

Response: As Reviewer suggested that this lignin stream should be characterized, “The compositional analysis was performed by the standard NREL laboratory analytical procedure (NREL/TP-510-42618). The ash content of the CEL was obtained gravimetrically after calcined in a muffle furnace at 575 °C. This purified CEL contained 90.6% acid-insoluble lignin, 3.8% acid-soluble lignin, 0.08% arabinose, 0.11% glucose, and 0.09% xylose. The ash and moisture contents of the raw lignin were 1.2% and 4.12%, respectively.” was added in the revised manuscript.

Pg 4, lines 150-151: is there really a difference between the V2 and V3 results? If the results shown in Figure 3 are claimed to be different, there should be some error bars on them. Also, what was the time and temperatures used to generate the data in Figure 3?

Response: As shown in figure 4, we believe that the conversion of lignin over V2 and V3 were almost constant but the product yield was reduced by V3. It is probably the CEL was partially over oxidized into carbon dioxide and water. The gas chromatograms from another side proved that there was a difference results between the V2 and V3, especially the product distribution (as shown in figure L2).

Figure L2. Gas chromatograms of silicotungvanadium polyoxometalates catalyzed depolymerization of CEL (reaction conditions: lignin 0.25g, catalysts 0.1 mmol, CH3OH/H2O=8:2 10ml, O2 2MPa, 150 °C, 3 h).

We repeated the experiment twice and added the error bars which represent standard deviations from three replicates. As to the reaction condition, “Reaction conditions: Lignin 0.25 g, catalysts 0.1 mmol, CH3OH/H2O=8:2 10 ml, O2 2 MPa, 150 °C, 3 h.” was also added in the figure caption. Similarly, the same reaction condition was added in figure 5 caption.

Figure 4: It would be helpful to explain that the figure is showing only THF-soluble products, indicating that the parent lignin is essentially insoluble in THF (presumably due to high MW), whereas the post-reaction oils are all significantly soluble. An indication of approximate MW in the figure would also be helpful.

Response: Considering the Reviewer’s suggestion, we have re-written this part as “As shown in figure 5, the parent lignin was essentially insoluble in THF (presumably due to high MW), whereas the post-reaction oils were all significantly soluble. GPC profiles of the CEL and depolymerized products catalysed by the different number of vanadium substituted POMs suggested the molecular weight was significantly decreased in average molecular weight (MW 1300 g mol-1) relative to the raw CEL (MW 4200 g mol-1) during the catalytic oxidative reactions.” Accordingly, we also labeled the approximate MW in the figure 5.

Figure 6 (first one): The reaction conditions that were held constant in these experiments should be denoted (e.g., what temperature was the left-side panel run at, and what time was the right-side panel run at?)

Response: It is really true as Reviewer suggested that the reaction conditions should be denoted, and “Reaction for left: Lignin 0.25 g, K6[SiV2W10O40] 0.1 mmol, CH3OH/H2O=8:2 10 ml, O2 2 MPa, 150 °C. Reaction for right: Lignin 0.25 g, K6[SiV2W10O40] 0.1 mmol, CH3OH/H2O=8:2 10 ml, O2 2 MPa, 3 h.” was added.

Table 1: The field has really moved beyond simple identification of products by GCMS. The authors need to provide the yield of monomeric products.

Response: It is really true as Reviewer suggested that we should provide the yield of monomeric products. We calculated the yield of the main phenolic compounds according the GC/MS, and added the results in Table 1.

Pg 8, line 225: it looks like the decrease in oil yield is about 10% absolute (more like 15-20% relative), so calling it a “stable” catalyst seems like a bit of a stretch. Some post-reaction catalyst characterization would help evaluate changes to the catalyst. Also, Figure 7 is labeled as Figure 6.

Response: Due to the complex structure of lignin, the results of recycling experiment were not easy to perform repetition. Herein, we repeated the experiment twice and added the error bars which represent standard deviations from three replicates.

Once again, thank you very much for your comments and suggestions.

Reviewer 2 Report

Generally, the studies are performed logically and the analytical techniques are used properly. It is very useful research and the novelty can be clearly seen.

Some minor comments are presented below:

Line 15: "Depolymerized products were observed by" should be rewritten to " Depolymerized products were analyzed by"

FTIR spectra of  K5[SiVW11O40], K6[SiV2W10O40], and K6H[SiV3W9O40: generally, the band from Si-O bond is strong and wide but maybe for these particular compounds it is small.

Lines 190-192: In my opinion, it can be in the error limit and the optimal time of depolymerization process is 2-3h.

Lines 208-215:  Vanillin is generally the main compound of the lignin depolymerization also in others known depolymerization processes of lignin.

Figure 5 and 6: The errors should be shown although I understand that for these results it is not easy to perform repetition.

Author Response

List of Responses

Dear Editors and Reviewers:

Thank you for your letter and for the reviewers’ comments concerning our manuscript entitled “Oxidative depolymerization of cellulolytic enzyme lignin over silicotungvanadium polyoxometalates” (ID: 443754). Those comments are all valuable and very helpful for revising and improving our paper, as well as the important guiding significance to our researches. We have studied comments carefully and have made correction which we hope meet with approval. Revised portion are marked using the “Track Changes” function in Microsoft Word. The main corrections in the paper and the responds to the reviewer’s comments are as flowing:

Line 15: "Depolymerized products were observed by" should be rewritten to " Depolymerized products were analyzed by"

Response: We have made correction according to the Reviewer’s comments. The statements of “observed” were rewritten as “analyzed”.

FTIR spectra of K5[SiVW11O40], K6[SiV2W10O40], and K6H[SiV3W9O40: generally, the band from Si-O bond is strong and wide but maybe for these particular compounds it is small.

Response: We used FTIR to identify the Keggin structural vibrations of POMs characteristic peaks at 1008 cm−1, 962 cm−1, 896 cm−1, and 18875px−1. As the reviewer said, maybe the Keggin structure affected the Si-O bond, which was smaller than other compounds in the FTIR spectra.

Lines 190-192: In my opinion, it can be in the error limit and the optimal time of depolymerization process is 2-3h.

Response: As we discussed in the manuscript: “When the reaction time prolonged from 2 h to 3 h, the lignin conversion yield and oily products yield changed from 74% and 48% to 87% and 53%, respectively.” We believed the optimal time of depolymerization process is 3h.

Lines 208-215:  Vanillin is generally the main compound of the lignin depolymerization also in others known depolymerization processes of lignin.

Response: We agree with the reviewer’s comment, and vanillin is one of the target products that we focus on.

Figure 5 and 6: The errors should be shown although I understand that for these results it is not easy to perform repetition.

Response: We repeated the experiment twice and added the error bars that represent standard deviations from three replicates.

Once again, thank you very much for your comments and suggestions.

Round 2

Reviewer 1 Report

I do not have much for new comments on this manuscript; the changes indicated in the authors' response letter satisfied my concerns. I feel it is now suitable for publication.